# NLI4CT: Multi-Evidence Natural Language Inference for Clinical Trial Reports

**Maël Jullien[1], Marco Valentino[3], Hannah Frost[1, 2], Paul O'Regan[2],**
**Donal Landers[2], André Freitas[1, 2, 3]**

[1]Department of Computer Science, University of Manchester, United Kingdom
[2]Digital Experimental Cancer Medicine Team, Cancer Research UK Manchester Institute
[3]Idiap Research Institute, Switzerland

## Abstract

How can we interpret and retrieve medical evidence to support clinical decisions? Clinical trial reports (CTR) amassed over the years contain indispensable information for the development of personalized medicine. However, it is practically infeasible to manually inspect over 400,000+ clinical trial reports in order to find the best evidence for experimental treatments. Natural Language Inference (NLI) offers a potential solution to this problem, by allowing the scalable computation of textual entailment. However, existing NLI models perform poorly on biomedical corpora, and previously published datasets fail to capture the full complexity of inference over CTRs.

In this work, we present a novel resource to advance research on NLI for reasoning on CTRs. The resource includes two main tasks. Firstly, to determine the inference relation between a natural language statement, and a CTR. Secondly, to retrieve supporting facts to justify the predicted relation. We provide NLI4CT, a corpus of 2400 statements and CTRs, annotated for these tasks. Baselines on this corpus expose the limitations of existing NLI approaches, with 6 state-of-the-art NLI models achieving a maximum F1 score of 0.627. To the best of our knowledge, we are the first to design a task that covers the interpretation of full CTRs. To encourage further work on this challenging dataset, we make the corpus, competition leaderboard, and website, available on CodaLab, and code to replicate the baseline experiments on GitHub[1].

## 1 Introduction

Clinical trials are research studies performed to test the efficacy and safety of novel treatments, and they are indispensable for the progression of experimental medicine (Avis et al., 2006). CTRs are documents that detail the methodology and results

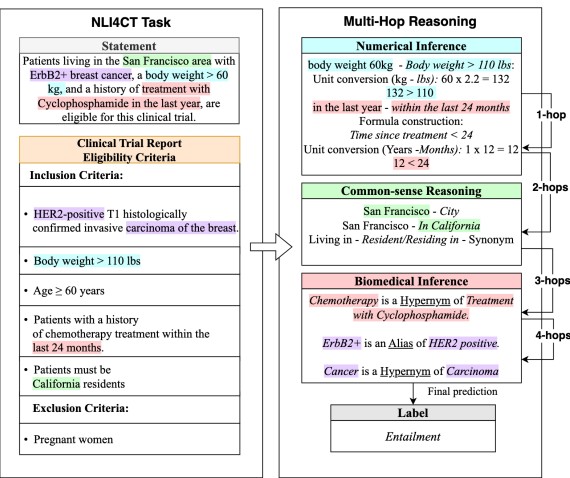

Figure 1: We propose two tasks for reasoning on clinical trial data expressed in natural language. Firstly, to predict the entailment of a **Statement** and a **CTR** premise, and secondly, to extract evidence to support the label.

of a particular trial. Clinical practitioners use the information in these CTRs to design personalized and targeted interventions, matching therapeutic agents to each patient's tumor biomarkers. However, there are over 400,000 CTRs available, being continually published at an ever-increasing rate (Bastian et al., 2010). Consequently, it has become unfeasible to manually carry out comprehensive evaluations of all the relevant literature when designing treatment protocols (DeYoung et al., 2020). Natural Language Inference (NLI) offers a potential solution for the large-scale interpretation and retrieval of medical evidence, connecting the latest evidence to support personalized care (Sutton et al., 2020).

In this paper, we propose two different tasks, on breast cancer CTRs. Firstly, to determine the inference relation between a natural language statement and a CTR, as shown in Figure 1. Secondly, to retrieve supporting facts from the CTR(s) to justify the predicted relation. With respect to healthcare, these inference tasks are a move towards automatic claim verification and evidence extraction, over

---

[1]email: `mael.jullien@manchester.ac.uk`

biomedical corpora.

These proposed tasks represent several challenges for existing NLI systems. The inference task requires substantial amounts of quantitative reasoning and numerical operations, as seen in Figure 1, on which many current state-of-the-art NLI models perform poorly (Ravichander et al., 2019; Galashov et al., 2019). Additionally, much of the inference for these tasks revolves around domain-specific terminology (see Figure 1). Many state-of-the-art NLI models fail to effectively surmount the word distribution shift from general domain corpora to biomedical corpora through transfer learning (Lee et al., 2019). This phenomenon is well-represented in the proposed dataset. This task differentiates itself from other biomedical NLI tasks by considering the full scope of CTRs, not limited to results and interventions, instead including eligibility and Adverse events (AE) as well (DeYoung et al., 2020). The proposed task also provides a significant variance in the types of inference required to solve each instance, with minimal, if any, repetition in inference chains. The contributions of this paper are as follows:

1. The definition of a novel benchmark (NLI4CT), including two main tasks requiring reasoning over CTRs that incorporate several of the fundamental challenges currently faced by modern NLI systems.

2. The release of a newly publicly-available corpus containing 2,400 expert-annotated entailment relations, with associated CTRs from ClinicalTrials.gov[2], labels, and extracted evidence lists.

3. An extensive empirical evaluation of state-of-the-art NLI models demonstrating current limitations and challenges involved in the proposed tasks. Specifically, we test 7 representative NLI models on our dataset and report a maximum F1 score of 0.644. We attribute these results to generalization problems induced by the shift in distribution required for inference.

To the best of our knowledge, we are the first to design a task that covers the full scope of CTRs, combining together the complexity of biomedical and numerical NLI. Solving these tasks would be a significant advancement toward efficient retrieval,

---

[2] https://clinicaltrials.gov/ct2/home

synthesis, and inference of published literature to support evidence-based experimental medicine.

## 2 Related Work

There are a multitude of expert-annotated resources for clinical NLP, examples are shown in Table 1. The TREC 2021 Clinical Track (Soboroff, 2021) is a large-scale information retrieval task on CTR data, with an emphasis on eligibility. Evidence Inference 2.0 (DeYoung et al., 2020) presents a Question-Answering (QA) task and a span selection task on CTR data, specifically with the CTR results. The MEDNLI (Romanov and Shivade, 2018) dataset contains an entailment task, which utilizes the medical history notes of patients as premises. These datasets are predominantly designed to test for biomedical language understanding and reasoning. Whereas NLI4CT, additionally requires complex numerical inference and common-sense reasoning to solve and is designed to avoid repetitive inference patterns. Currently, neural architectures achieve the best results on biomedical NLI datasets (Gu et al., 2021; DeYoung et al., 2020).

Neural models continue to perform poorly on quantitative reasoning and numerical operations within NLI (Ravichander et al., 2019; Galashov et al., 2019). Prior works such as Lee et al. (2020), Shin et al. (2020), and Gu et al. (2021) experiment with biomedical pre-training strategies. Kiritchenko et al. (2010) presents ExaCT, an automatic information extraction system for clinical trial variables such as drug dosage, sample size, and primary outcomes from CTRS. While there are many proficient evidence retrieval models, there are no existing systems that are able to effectively carry out biomedical and numerical NLI simultaneously.

In a related vein, Hao et al. (2014) introduced an automated approach to cluster clinical trials based on similar eligibility features. Using data from ClinicalTrials.gov, they extracted semantic features from the eligibility criteria of trials and employed the Jaccard similarity coefficient for clustering. Their method, which generated 8,806 center-based clusters, offers potential utility in understanding knowledge reuse patterns in clinical trial designs and enhancing recruitment strategies. Liu et al. (2021) presented the Clinical Trial Knowledge Base (CTKB), a comprehensive database that houses discrete clinical trial eligibility criteria. Developed using advanced NLP techniques, CTKB transforms free-text criteria from ClinicalTrials.gov

| Dataset | Task | Metrics | Dataset size | Labels |
|---------|------|---------|--------------|--------|
| NLI4CT (2023) | Predict the entailment of a statement and one of 4 sections of a clinical trial and select evidence to support the label. | Macro F1 mAP | 2400 instances | Entailment Contradiction |
| Evidence Inference 2.0 (DeYoung et al., 2020) | Given a treatment A, a comparator B, an outcome, and a CTR, predict the relationship between A and B relative to the outcome. Then identify spans of evidence in the CTR. | Macro F1 Evidence token AUC | 10,000+ QA pairs | significantly increased significantly decreased |
| MEDNLI (Romanov and Shivade, 2018) | Given an annotated sentence, and a sentence from patient medical records determine the entailment relation. | Accuracy | 14,000+ sentence pairs | Entailment neutral Contradiction |
| TREC clinical track (Roberts et al., 2021) | Given synthetic patient descriptions/clinical notes retrieve relevant CTRs for which this patient would be eligible. | NDCG Precision | 75 synthetic patient cases 350K clinical trials | Eligible Excludes Not Relevant |

Table 1: Description of the main Clinical NLP datasets present in the literature.

into structured concepts and attributes, facilitating improved querying and analysis. This knowledge base holds promise in enhancing clinical trial cohort definitions, assessing population representativeness, and bridging data gaps in electronic health records for clinical trial recruitment. In a complementary effort to streamline clinical trial recruitment Miotto and Weng (2015), introduced a case-based reasoning approach using electronic health records (EHRs). By aggregating the EHR data of a minimal sample of trial participants, they created a "target patient" profile for each trial. This profile was then used to efficiently identify new eligible patients based on their relevance to the "target patient." Their method, evaluated on 13 diversified clinical trials at Columbia University, demonstrated high precision in identifying eligible patients, emphasizing the capability of EHR-driven solutions in streamlining clinical trial recruitment.

As the biomedical domain continues to generate vast amounts of textual data, the importance of efficient concept recognition grows. Kors et al. (2015) introduced the Mantra GSC, a multilingual gold-standard corpus designed for biomedical concept recognition across five languages: English, French, German, Spanish, and Dutch. This corpus, based on parallel biomedical texts from sources like Medline and drug labels, provides annotations grounded in a subset of the Unified Medical Language System. Such endeavors underscore the global push towards more multilingual biomedical information extraction tools, catering to the diverse linguistic landscape of the biomedical community. In the pursuit of enhancing access to evidence-based medicine, Campillos-Llanos et al. (2021) introduced the Clinical Trials for Evidence-Based Medicine in Spanish (CT-EBM-SP) corpus. This corpus, comprising 1200 texts about clinical trials, is annotated with entities from the Unified Medical Language System, covering areas such as anatomy, pharmacological substances, pathologies, and diagnostic or therapeutic procedures. Their work emphasizes the potential of natural language processing in bridging the gap between vast medical literature and the need for healthcare professionals to access relevant, evidence-based information efficiently.

## 3 Clinical Trial Reports

Clinical trials are research studies on human volunteers used to evaluate an intervention (DeYoung et al., 2020). An intervention may be medical, surgical, or a behavioral protocol. These interventions are often compared with a standard treatment, placebo, or control protocol. The investigators use various outcome measurements from the volunteers to ascertain the effectiveness of the intervention, as well as the likelihood and severity of AEs.

For NLI4CT we retrieved 1000 publicly available Breast cancer CTRs, with published results from ClinicalTrials.gov, on the 22nd of December 2021. This data is developed by the U.S. National Library of Medicine and covered by the HIPAA Privacy Rule, protecting personally identifiable information (additional ethical considerations can be found in Section 11). We separate the CTRs into 4 sections:

**Eligibility criteria:** A set of conditions for patients to be allowed to take part in the clinical trial.

**Intervention:** The type, dosage, frequency, and duration of treatments being studied.

**Results:** Number of participants in the trial, outcome measures, units, and the results.

**Adverse Events:** These are signs and symptoms

| Eligibility Criteria | **Inclusion:**
• HER2 + diagnosis
• 18 years of age or older
**Exclusion:**
• Presence of central nervous system or brain metastases. |
|---|---|
| **Intervention** | **Gemcitabine** 1500 mg/m2 body surface area (BSA) **intravenously** (IV) over 30 minutes (+/- 5 minutes) on days 1 and 15 of each cycle. |
| **Results** | **Variable:** Progression Free Survival
Number of Participants: 29
Median (months): 10.4 (5.6 to 15.2) |
| **Adverse Events** | **Total:** 3/29
• Neutropenic Fever 1 / 29 (3.45%)
• Peripheral Neuropathy 1 / 29 (3.45%)
• Seizure/Syncope 1 / 29 (3.45%) |

Table 2: Sample excerpts from each section in a CTR. Data provided by ClinicalTrials.gov.

observed in patients during the clinical trial.

See examples of the types of information contained in these sections in Table 2.

## 4 Task Definition

We define two different tasks on the NLI4CT dataset, Task 1, a textual entailment task, and Task 2, an evidence selection task. Each instance in NLI4CT contains a CTR premise and a statement. The CTR premise comprises one of the four sections of a CTR, with both a mean token length and a standard deviation of 230. The statements are sentences ranging from 10-35 tokens (see example in Figure 1). On average there are 7.74 pieces of relevant evidence that need to be selected out of a total of 21.67 facts per CTR premise. There are two types of instances in NLI4CT, in *single* type instances the CTR premise is the primary CTR, and the statement makes a claim about the information in this CTR. Whereas in the *comparison* type the premise contains a primary and a secondary trial, and the statement will make a claim comparing and contrasting the two trials. To summarize:

**Task 1** Requires determining the entailment relation between the CTR premise and the statement, therefore the output is either an entailment or contradiction label, as shown in Figure 1.

**Task 2** Requires outputting a set of supporting facts, extracted from the CTR premise, necessary to justify the label predicted in Task 1.

## 5 Annotation

### 5.1 Statement annotation

A group of 4 domain experts, including clinical trial organizers from the Manchester Cancer Institute and the Digital Experimental Cancer Medicine Team (DECMT), took part in the annotation task, with one expert performing the majority of the annotations. We supplied annotators with an annotation guide, displayed in the appendix, which was refined following the annotation of 100 pilot examples by the principal expert annotator to optimize the overall annotation process. Annotators were provided with a CTR section prompt, and two CTRs, a primary and a secondary, as seen in step (B) of Figure 2. The annotation task was first to generate an entailment statement, a short statement that makes an objectively true claim about the contents of the prompted section of the trial(s). In step (C) of Figure 2 annotators could choose to write a statement about the contents of the primary trial or to compare both the primary and secondary trials. The objective was to generate non-trivial statements, encouraging to solve the inference by reading and understanding multiple rows of the CTR section. Non-trivial statements typically included summarisation, comparisons, negation, relations, inclusion, superlatives, aggregations, or rephrasing. However, the sentences were not limited to these types, and any statement involving understanding and reasoning was accepted.

Each CTR is divided into 4 sections (Table 1), and each of these sections represents a set of facts. And for all annotated statements we select a subset of these facts that support the label assigned to the statement, see Figure 1 and (D) in Figure 2. This collection of facts is designated as evidence. There is always at least one piece of evidence relevant to the inference for any given statement. In cases where negation is used, e.g. *clinical trial A does not have a placebo arm* annotators are asked to provide the full CTR section as evidence, as we believe this type of retrieval is more reflective of human inference patterns.

Finally, we employ a negative rewriting strategy (Chen et al., 2019), using the previously generated entailment statement to write an objectively wrong contradiction statement, as shown in Step (E) of Figure 2. Annotators are encouraged to modify the words, phrases, or sentence structures while retaining the sentence style/length of the original statement. This is done to minimize the occurrence

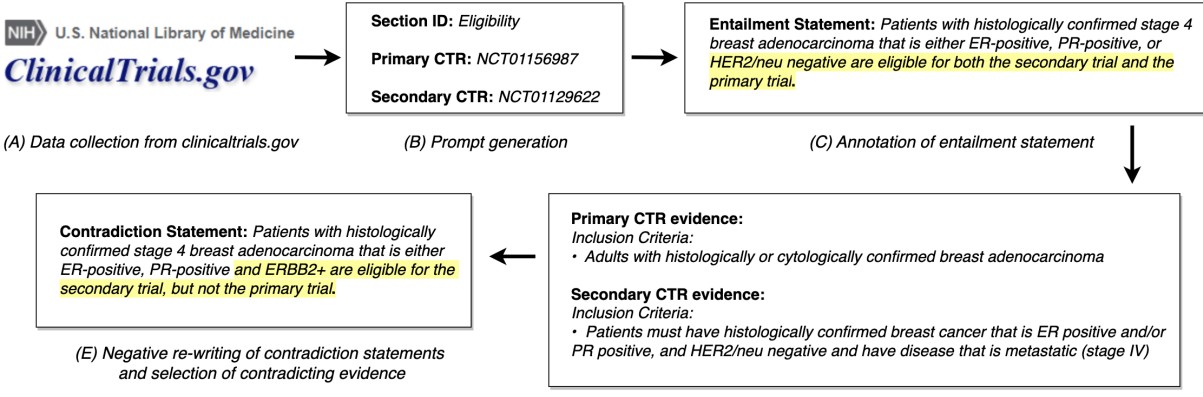

Figure 2: Overview of the NLI4CT annotation process.

of stylistic or linguistic patterns pertaining to either entailment or contradictory statements. We then collect evidence that contradicts the claims made in the contradiction statement. An example annotation form is available in the appendix.

For quality assurance, 3 volunteers re-classified 100 randomly sampled test instances, recording an average accuracy of 85% with respect to our gold labels and a Cohen's kappa of 0.83, indicating substantial inter-annotator agreement.

## 5.2 Resulting Dataset

The NLI4CT dataset consists of 2,400 annotated statements with accompanying labels, CTRs, and evidence. Split into 1700 training, 500 test, and 200 development instances. The two labels and 4 CTR sections prompts are equally distributed across the dataset and its splits. 60% of the instances are the *single* type, with the remaining 40% the *comparison* type. The total annotation effort for NLI4CT accumulated to approximately 260 person-hours.

## 6 Reasoning Challenges

### 6.1 Biomedical Reasoning

**Acronyms.** In CTRs acronyms are used for concepts such as test names, administration times and orders, treatment types, and disease names. Acronyms are significantly more prevalent in clinical texts than in general-domain texts and consistently disrupt NLP performance in this field (Grossman Liu et al., 2021; Shickel et al., 2017; Jiang et al., 2011; Moon et al., 2015; Jimeno-Yepes et al., 2011; Pesaranghader et al., 2019; Jin et al., 2019; Wu et al., 2015). Over 100,000 healthcare acronyms and abbreviations have been identified with 170,00 corresponding senses (Grossman Liu

et al., 2021). Future systems must overcome this challenge to perform biomedical NLI.

**Synonyms and Aliases.** These are often employed for drugs and gene names. Statements in NLI4CT regularly refer to treatments and diagnoses using different aliases than are present in the CTR premise, e.g. the medication Trastuzumab has 6 different brand names, and the gene ERBB2 has over 20 aliases.

**Taxonomic Relations.** Concepts such as diseases, treatments, and diagnostic tests can be classified and structured in a taxonomic hierarchy. As shown in Figure 1, to achieve the necessary inference systems must understand that chemotherapy is a hypernym or a super-set of Cyclophosphamide treatments, similar to Cancer being a hypernym of Carcinoma.

**Domain Knowledge.** NLI4CT statements regularly test for domain expert knowledge. This is done by describing characteristics or conditions which pertain to certain biomedical grading systems or diagnoses. For example, if a patient is confined to bed or a chair for more than 50% of waking hours, and the inclusion criteria require a WHO score of < 2, the model must be able to infer that the patient will not be eligible. Or if a patient has a positive FISH test, and the inclusion criteria require the patient to have a Triple Negative Breast Cancer diagnosis, they would also be ineligible.

### 6.2 Commonsense Reasoning

NLI4CT requires common sense reasoning, particularly co-reference resolution and general world knowledge, which remain challenging for existing NLI models (Emami et al., 2018; Trichelair et al.,

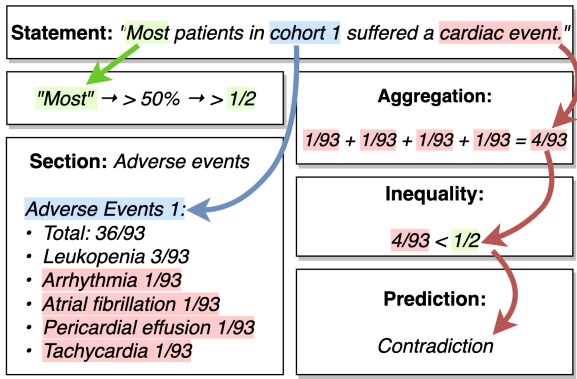

Figure 3: Schematic of a **numerical inference** chain to solve an NLI4CT instance.

2018). Co-referencing resolution is often necessary to associate dosages, frequencies, and routes of administration to specific drugs in the intervention. General world knowledge is most often required for claims about the eligibility section, e.g. a child is a person < 18 years old, and women over the age of 60 are not of childbearing potential.

### 6.3 Numerical Reasoning

The type of numerical reasoning required to solve instances most closely resembles Math Word Problems (MWP) (Huang et al., 2016; Patel et al., 2021). Prior works have shown that even elementary MWPs remain unsolved for current NLP models (Patel et al., 2021). NLI4CT requires models to compare dosages, frequencies, and fractions/percentages often necessitating unit conversions. Consider the example instance in Figure 3. Here the model will need to aggregate the number of occurrences of every different cardiac event recorded in the adverse events of cohort 1. Then it must construct and evaluate the inequality. This combination of biomedical knowledge and numerical reasoning is expected to be a significant challenge for existing models.

## 7 Experiments

### 7.1 Baseline Model Architectures

We establish a baseline performance on NLI4CT, to assess the strengths and limitations of existing NLI models. We test a range of transformer-based architectures; BERT-base (Devlin et al., 2018), RoBERTa-base (Liu et al., 2019), GPT-2 (Radford et al., 2019), T5-base (Raffel et al., 2020), Megatron-lm (Shoeybi et al., 2019), and GPT 3.5 turbo (Brown et al., 2020). We chose GPT-3.5 Turbo because it offers a more cost-effective solution than GPT-4 and consistently demonstrates superior performance compared to LLaMa. Additionally, we experiment with the bag-of-words ranking function Okapi BM25 (Fang et al., 2004; Robertson et al., 1995).

### 7.2 Biomedical Pretraining

Transformer-based models leverage pre-trained contextualized word embeddings to model inference, and optimizing pre-training strategies can drastically improve downstream performance (Gururangan et al., 2020). Previous works have repeatedly shown that pre-training general-domain language models on biomedical corpora significantly improves performance on bio-medical tasks (Shin et al., 2020; Beltagy et al., 2019; Lee et al., 2020; Li et al., 2016). Therefore, we also include BioBERT (Lee et al., 2020), and BioMegatron (Shin et al., 2020), versions of BERT-base and Megatron-lm that are pre-trained on biomedical corpora, in our baselines to assess the effects of pre-training on performance on our dataset. We selected these models as emblematic of the domain-specific model research community, deferring a detailed evaluation of such models for future studies.

### 7.3 Experimental Details

For the transformer-based model baselines we tokenize the statements and the section(s) of the CTR(s) as indicated by the instance type and section prompt. If the instance type is *comparison* we concatenate the primary and secondary CTR section data together. Then we pass the tokenized statement and section information to the model. It should be noted that for a significant portion of the instances, particularly of the *comparison* type, the input exceeds the maximum input length of 512 tokens. We employ a binary sequence classification/regression head for all baseline models, which predicts either *Entailment* or *Contradiction*. The models were trained on the training set for 10 epochs, and each model checkpoint was archived. We additionally test zero-shot GPT 3.5 turbo and provide the prompt in the appendix. Finally, in line with previous work on explainable NLI (Valentino et al., 2022a,b; Jansen and Ustalov, 2019; Thayaparan et al., 2021) we adopt BM25 to derive sparse vectorial sentence representations, and subsequently, employ cosine similarity as a bounded scoring function then apply a threshold on the cosine similarity between statements and

| Model | F1 | Precision | Recall | Accuracy |
|-------|-----|-----------|--------|----------|
| BM25 | 0.627 | 0.490 | **0.872** | 0.482 |
| BERT-base | 0.528 | 0.555 | 0.504 | 0.550 |
| RoBERTa-base | 0.628 | **0.654** | 0.604 | **0.642** |
| BioBERT | **0.644** | 0.633 | 0.656 | 0.638 |
| BioMegatron | **0.644** | 0.585 | 0.716 | 0.604 |
| GPT2 | 0.603 | 0.500 | 0.760 | 0.500 |
| T5-base | 0.602 | 0.504 | 0.748 | 0.506 |
| GPT 3.5 turbo | 0.604 | 0.630 | 0.580 | 0.620 |

Table 3: Results from the NLI4CT **Task 1** baselines on the test set.

premises to perform binary classification (i.e., predicting entailment if the cosine similarity is above the threshold, contradiction otherwise), we test threshold values from 0.1 - 0.9. We then record the results of these models on the test set. Code to reproduce the experiments is available on GitHub.

## 8 Evaluation Metrics

Task 1 is a binary classification task, the statement is either labeled as *entailment* or *contradiction*. We evaluate performance on task 1 by computing the Precision, Recall, and Macro F1-score of the predicted labels against the annotated gold labels (Entailment/Contradiction). For Task 2 the output is a subset of the facts within the CTR premise. We can frame the evidence selection problem as a ranking problem (Jansen et al., 2021). Models assign a score to each fact in a CTR premise, which can be used to rank the facts. In this framework, solving the task requires the model to rank all the facts in the gold evidence higher than the irrelevant facts. We evaluate performance on this task using Mean Average Precision mAP @ K (Liu, 2011).

## 9 Experiment Results

### 9.1 Task 1

Table 3 summarises the performance of the baseline models on Task 1. We report the results from the model checkpoints with the highest F1 score on the test set. BM25, T5, BERT-base, and GPT2 fail to achieve over 0.5 accuracy on the test set. The best-performing models on the test set were BioBERT, BioMegatron, and RoBERTa-base, each achieving over 0.6 accuracy and F1 score. BERT base reported a significantly lower F1 than the other baselines.

BioBERT, and BioMegatron both achieve over 0.95 accuracy and F1 score on the training set after 10 epochs. As shown in Table 3, this does not translate to an improvement in performance on the test, indicating drastic over-fitting. BioBERT significantly outperforms its general-domain counterpart on the training set, achieving a maximum F1 score of 0.96, in contrast to 0.65 by BERT-base. Biomedical pre-training clearly impacts the models' ability to encode relevant information, limited to the training set. On average, model complexity does not have a consistent impact on performance. Neither the over-fitting biomedical models nor the models which failed to learn on the training set exhibited any positive responses to changes in the learning rate. The aggregation of these results indicates that the NLI4CT entailment task represents a significant challenge for existing NLI models.

### 9.2 Evidence-Only Baseline

We generated an additional set of baselines, using only the gold evidence as the premise. This reduces the likelihood of the models focusing on irrelevant or adversarial information and removes the need to identify evidence spans within the CTR section. This did not result in any significant differences in performance, with BioBERT achieving the highest F1 score of 0.667. This implies that even after training for 10 epochs none of the models effectively draw inferences from the relevant evidence. The results for this baseline are available in the appendix.

### 9.3 Statistical Artefacts

We test BERT-base, using only the statements, removing any access to the CTRs. BERT-base achieved a maximum F1 score of 0.626 and an accuracy of 0.606 on the test set, a significantly higher F1 score than the BERT-base Task 1 baseline (0.528). Additionally, we observed above-random accuracy (0.5) with several model checkpoints. This indicates that the evaluated models exclusively rely on the presence of superficial statistical artefacts without learning the underlying rules of the tasks. Lexical and syntactic patterns such as token distributions, statement lengths, and discriminative conditions that are disproportionately associated with a particular class of a dataset can superficially inflate model performance (Herlihy and Rudinger, 2021). These statistical artefacts are typically introduced during the annotation (Gururangan et al., 2018) and allow statement-only

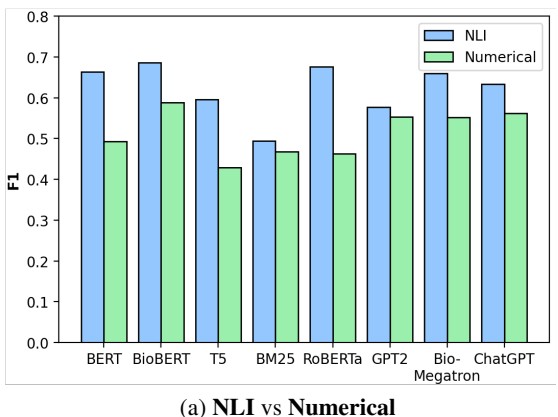

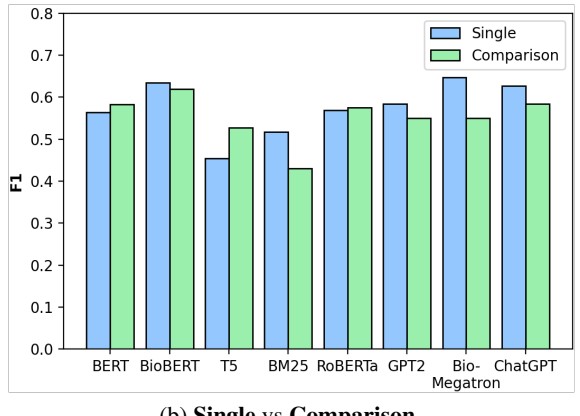

(a) **NLI** vs **Numerical**    (b) **Single** vs **Comparison**

Figure 4: F1 score achieved by the baseline models on different test splits.

classifiers to achieve significantly above-random results (Poliak et al., 2018; Tsuchiya, 2018). We support the statement-only baseline with qualitative analysis of the dataset. We found that there was no significant difference between the sentence length distributions of the two classes. Additionally, we find a 100% overlap in the 15 most common tokens between the two classes in the training set and 93% overlap in the test set. However, several individual tokens were not evenly distributed across classes in the test set. In particular the distributions of *not, more, group, and cohort*. However, these uneven distributions were not present, if not completely reversed in the training set. Therefore these do not entirely explain the above random accuracy.

We conducted an in-depth investigation into potential artefacts and biases throughout our experiments, yielding no definitive evidence of substantial statistical biases. Specifically, the statement-only baseline attains an F1 score of 0.626, notably below the majority class baseline, which is 0.66 F1. Given that the entailment task involves binary classification and acknowledging that existing models have a propensity for identifying spurious correlations in data, it can be concluded that the dataset contains minimal such artefacts.

### 9.4 Categorical Analysis

**Numerical Reasoning** We categorize the instances in our dataset into those that require numerical or quantitative reasoning and those that require pure NLI. We separate out all instances with statements that contain numbers or any of the following tokens; *higher, lower, fewer, more, less, number, same* into the Numerical reasoning category. This results in a 62% Numerical 38% NLI split

of the test set. We evaluate the best-performing checkpoints on these categories and report the results in Figure 4. BioBERT reported the highest F1 score for both Numerical and NLI categories, at 0.588 and 0.686 respectively. The performance on the NLI instances is significantly higher than the performance on the numerical instances, for all models other than BM25 and GPT2 which only marginally report the same trend. Biomedical pre-training does not appear to provide a significant advantage within either category. Overall these results follow the trends observed in prior works, numerical reasoning remains a challenging task for NLI models (Ravichander et al., 2019; Galashov et al., 2019; Peng et al., 2021; Patel et al., 2021).

**Single vs Comparison.** We repeat this experiment, this time categorizing by instance type. *Comparison* type instances have 2 CTR sections as a premise, effectively doubling the amount of irrelevant and potentially adversarial information. Therefore, we would expect *comparison* instances to be more challenging than the *single* type. We report the results of this experiment in Figure 4. Contrary to expectations the majority of the models there was no significant difference in performance across the categories. BioMegatron reports a +0.1 F1 on *single* type instances, and BM25, and T5 report +0.09 and +0.07 F1 score respectively on the *comparison* type. As T5 failed to learn Task 1, even on the training set, and BM25 is not able to capture clues that go beyond lexical overlaps, we hypothesize that this is in part due to the model relying on superficial clues to predict the output label. Therefore, in many cases, the evidence is just not being leveraged properly.

| | | **F1 score** | | |
|---|---|---|---|---|
| **Model** | AEs | Eligibility | Intervention | Results |
| BERT-base | 0.649 | 0.554 | 0.547 | 0.537 |
| BioBERT | 0.619 | **0.667** | 0.588 | **0.628** |
| BioMegatron | 0.661 | 0.576 | **0.626** | 0.528 |
| BM25 | 0.158 | 0.490 | 0.529 | 0.574 |
| GPT2 | 0.615 | 0.531 | 0.523 | 0.598 |
| T5-base | 0.504 | 0.515 | 0.500 | 0.455 |
| RoBERTa | 0.667 | 0.609 | 0.523 | 0.460 |
| GPT 3.5 turbo | **0.700** | 0.602 | 0.511 | 0.612 |
| Average Rank | 1.86 | 2.29 | 2.86 | 3.00 |

Table 4: F1 score of the baseline models on the test set, categorized by **CTR section**.

| Model | mAP |
|---|---|
| BM25 | **0.786** |
| all-MiniLM-L6-v2 | 0.777 |
| all-distilroberta-v1 | 0.762 |
| all-mpnet-base-v2 | 0.760 |
| sentence-BioBert | 0.752 |
| sentence-t5-base | 0.749 |

Table 5: Results from the NLI4CT **Task 2** baselines on the test set.

**CTR Sections.** Finally, we categorize the instances in our dataset by CTR section and report the results in Table 4. All models show significant fluctuations in performance across the different sections. Interestingly, Results and AEs are on opposite ends of the rankings, despite arguably containing the most similar types of information. Additionally, Eligibility is ranked $2^{nd}$ despite having the highest average number of tokens.

### 9.5 Task 2

For Task 2, the evidence selection task, we generate a baseline using SOTA general purpose transformer-based ranking models (Reimers, 2020), namely, DistilRoBERTa (Sanh et al., 2019), Mpnet (Song et al., 2020), and MiniLM (Wang et al., 2020). We additionally test sentence-t5-base (Ni et al., 2021), sentence-BioBERT (Deka and Jurek-Loughrey, 2021), and BM25. We embed the facts in the CTR sections, and the statements using these models and compute the cosine similarity. We evaluate using the mAP @ K (Liu, 2011), with K = *total number of sentences*, and report the results in Table 5. There is no significant difference between the mAP scores of the models, with BM25 producing the highest mAP score, indicating that task 2 is still challenging for dense retrieval models, even with biomedical pre-training.

### 10 Conclusions & Future work

We present 2 tasks, textual entailment of claims over clinical trial reports (CTRs), and extracting evidence from CTRs to support or contradict these claims. We provide a corpus of 2400 expert-annotated instances to test and train models on these tasks. If models could be developed to solve these tasks, it would progress the field toward the efficient retrieval and synthesis of published literature to support evidence-based medicine. Additionally, it would address several important NLU challenges related to numerical reasoning, biomedical NLI, and inference over long texts.

Our baselines outline the weaknesses of existing NLI models, particularly with regard to numerical inference. We tested 6 SOTA NLI models on Task 1 and reported a maximum F1 score of 0.644. Additionally, we show that the limiting factor for current models is not evidence selection, but rather leveraging that evidence, and using numerical and biomedical inference for predictions.

In future works, we plan to generate a subset of adversarial instances via a controlled set of perturbations of words/numbers in statements. Allowing us to perform interventional and causal analysis of models' behavior for numerical and multi-hop inference (Stolfo et al., 2022), in the context of biomedical applications, as well as to extend the size of the dataset. Our corpus, competition leaderboard, code, and website are available on CodaLab and GitHub.

### 11 Ethical Statement

**Annotation.** The domain expert annotators are all authors of this paper and were therefore not compensated. The three volunteers for the reclassification were recruited through personal referrals and contributed their efforts on a pro bono basis.

**Data Usage.** To the best of our knowledge, we are entirely in line with the ClinicalTrials.gov terms of service. We acknowledge ClinicalTrials.gov as the data source and explicitly specify the date of data retrieval. Additionally, we provide a detailed account of the modifications made to the dataset.

## 12 Limitations

The most significant limitation of this work is that despite the fact that we are dialoguing with a motivational scenario for medicine, these models are not fit for medical application, as this would require a technology clinical trial, generalization studies and regulatory assessment. Despite this, we believe that developing benchmarks such as NLI4CT represents a necessary step toward the development of models capable of clinical decision-making.

Additionally, our evaluation does not include an interventional study, meaning we do not perform perturbations/interventions on the test instances to verify that the models learn the underlying causal structure of the tasks. We plan to extend NLI4CT with interventional studies in future work.

Finally, our dataset contains fewer instances than other published biomedical NLI datasets. This is a consequence of the time and resource-intensive nature of expert-level annotation. The size of the training set may be problematic for large models.

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

## 12.1 Appendices

### 12.1.1 Pre-processing details

We extracted all CTR data from ClinicalTrials.gov on the $22^{nd}$ of December 2022. The Eligibility criteria and intervention information is retrieved from the **Eligibility criteria** and **Intervention** sections. The results information is taken from the **Primary Outcome** section. The Adverse events section is taken from the **Serious Adverse Events** section, not to be confused with the adverse events section. We remove or replace symbols or tags produced in the extraction, and segment the CTR data into individual lines. These are the only changes made to the CTR data.

### 12.1.2 Zero-shot GPT 3.5 prompt

prompt = *"Given a section of 2 clinical trial descriptions and a statement, determine whether the statement logically follows from the sections. If the statement logically follows from the sections, you need to return 'Entailment'. If the statement does not logically follow from the sections, you need to return 'Contradiction'. The output should be a single word <Entailment> or <Contradiction>. "*
*"Statement: "* + Statement
*"Primary Trial: "* + Primary CTR text
*"Secondary Trial: "* + Secondary CTR text

### 12.2 Tables and Graphs

| | F1 score | |
|---|---|---|
| **Model** | Single | Comparison |
| BERT-base | 0.564 | 0.582 |
| BioBERT | 0.635 | 0.619 |
| BioMegatron | 0.648 | 0.550 |
| BM25 | 0.517 | 0.431 |
| GPT-2 | 0.585 | 0.550 |
| RoBERTa-base | 0.569 | 0.576 |
| T5-base | 0.455 | 0.527 |
| GPT 3.5 turbo | 0.627 | 0.584 |

Table 6: F1 score of the baseline models on the test set, categorized by instance **type**.

| | F1 score | |
|---|---|---|
| **Model** | Numerical | NLI |
| BERT-base | 0.493 | 0.664 |
| BioBERT | 0.588 | 0.686 |
| BioMegatron | 0.552 | 0.660 |
| BM25 | 0.468 | 0.495 |
| GPT-2 | 0.553 | 0.577 |
| RoBERTa-base | 0.463 | 0.676 |
| T5-base | 0.429 | 0.596 |
| GPT 3.5 turbo | 0.562 | 0.635 |

Table 7: F1 score of the baseline models on the test set, categorized by **numerical** and **NLI** inference.

| **Model** | F1 | Precision | Recall | Accuracy |
|---|---|---|---|---|
| BM25 | 0.622 | 0.489 | 0.856 | 0.480 |
| BERT-base | 0.597 | 0.569 | 0.628 | 0.576 |
| RoBERTa-base | 0.659 | 0.506 | 0.944 | 0.512 |
| BioBERT | 0.667 | 0.589 | 0.768 | 0.616 |
| BioMegatron | 0.596 | 0.543 | 0.660 | 0.552 |
| GPT2 | 0.581 | 0.519 | 0.660 | 0.524 |
| T5-base | 0.521 | 0.522 | 0.520 | 0.522 |

Table 8: Results from the NLI4CT Task 1 **evidence-only** baseline on the test set.

| **Epoch** | F1 score | Accuracy |
|---|---|---|
| Pre-trained | 0.008 | 0.500 |
| 1 | 0.666 | 0.506 |
| 2 | 0.527 | 0.590 |
| 3 | 0.623 | 0.572 |
| 4 | 0.613 | 0.604 |
| 5 | 0.522 | 0.578 |
| 6 | 0.485 | 0.584 |
| 7 | 0.572 | 0.590 |
| 8 | 0.575 | 0.598 |
| 9 | 0.494 | 0.590 |
| 10 | 0.626 | 0.606 |

Table 9: BERT-base results on the **Statement-only** test set.

| model | learning rate | batch size | hidden_size | num_hidden_layers | num_attention_heads | link |
|---|---|---|---|---|---|---|
| BERT-base-uncased | 1.00E-05 | 32 | 768 | 12 | 12 | https://huggingface.co/bert-base-uncased |
| RoBERTa-base | 1.00E-05 | 32 | 768 | 12 | 12 | https://huggingface.co/roberta-base |
| BioBERT-v1.1 | 1.00E-05 | 32 | 768 | 12 | 12 | https://huggingface.co/dmis-lab/biobert-v1.1 |
| BioMegatron | 1.00E-05 | 16 | 1024 | 24 | 16 | https://catalog.ngc.nvidia.com/orgs/nvidia/models/biomegatron345muncased |
| GPT2 | 1.00E-05 | 32 | 768 | 12 | 12 | https://huggingface.co/gpt2 |
| T5-base | 1.00E-05 | 32 | 768 | 12 | 12 | https://huggingface.co/t5-base |

Table 6: Additional model implementation details for Task 1.

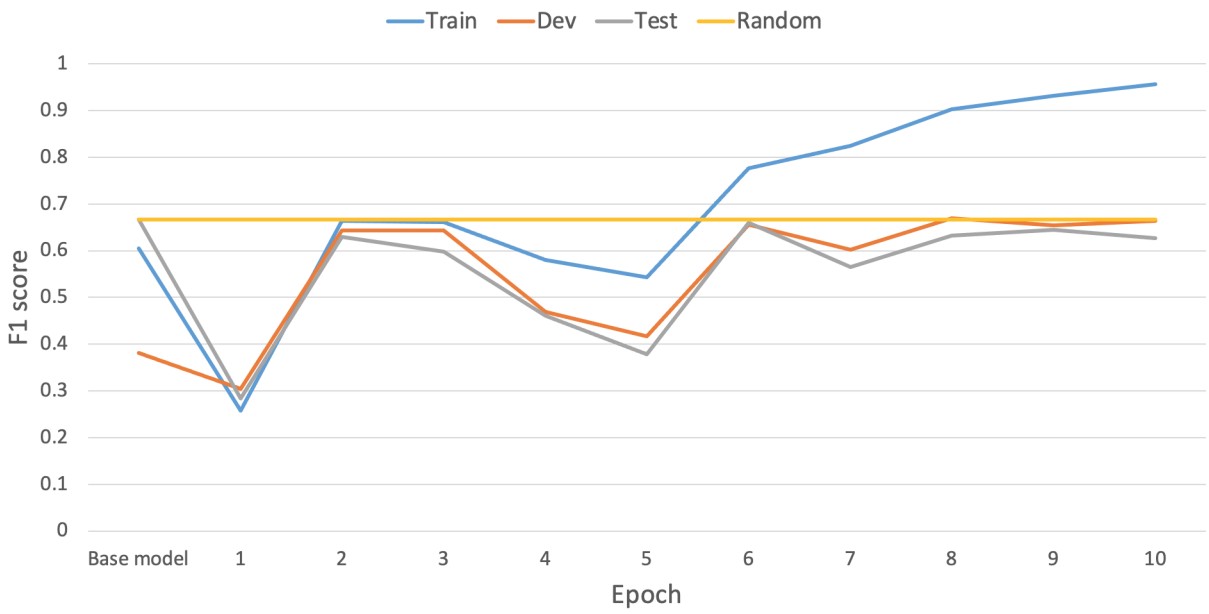

Figure 5: Graph of the **BioBERT** F1 score by epoch, on the test, training, and development set.

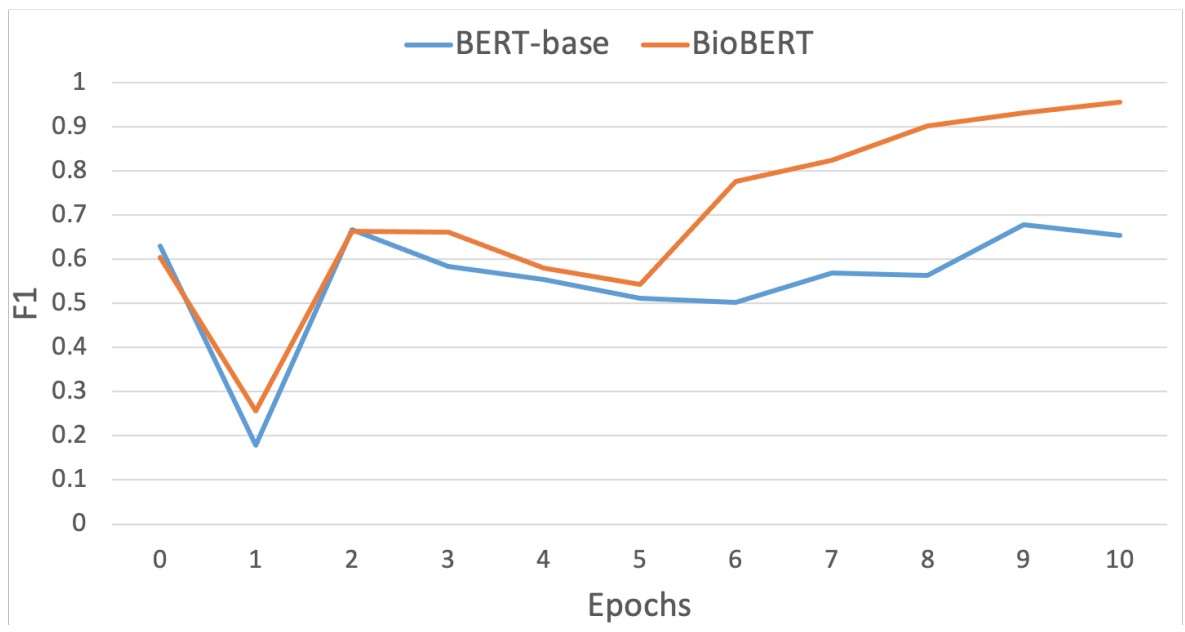

Figure 6: Graph of **BERT-base** F1 score by epoch, on the test set, compared to **Bio-BERT**.

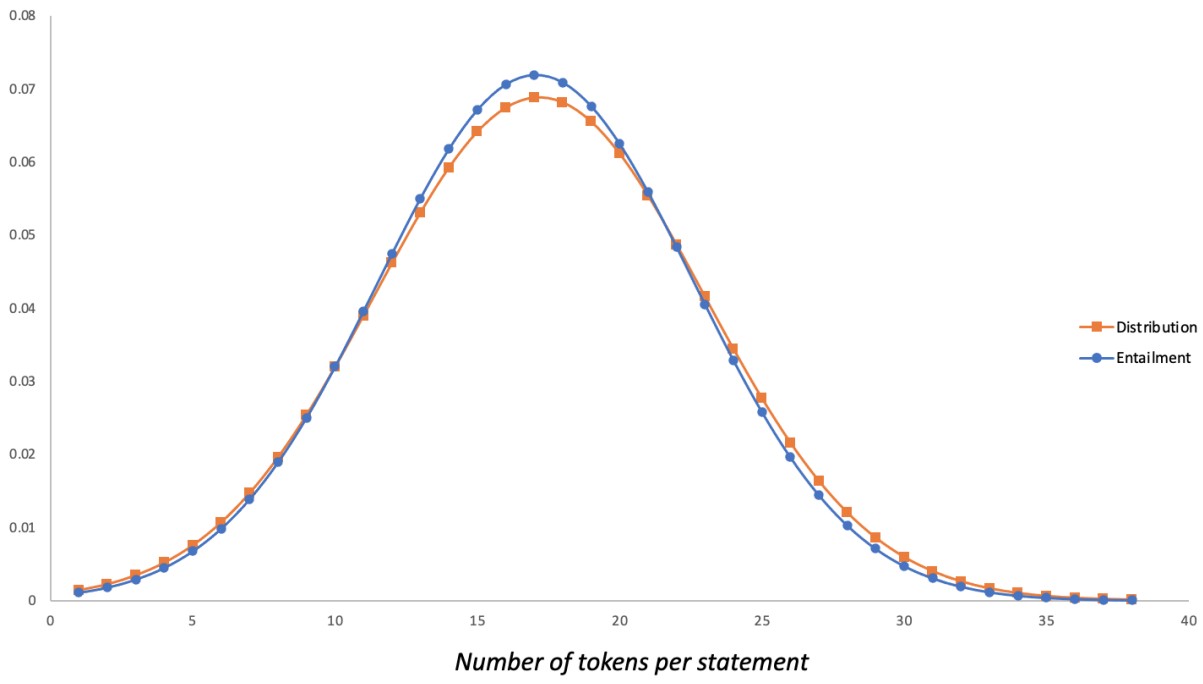

Figure 7: Distribution curve of the **number of tokens per statement** in the test and training set.

# Intervention

**Primary Clinical trial ID:**

NCT02953860

INTERVENTION 1:
- Fulvestrant With Enzalutamide
- 500mg of Fulvestrant will be given IM on days 1, 15, 28, then every 4 weeks as per standard of care (SOC) and 160mg of Enzalutamide will be given, in conjunction with Fulvestrant, PO daily.
- Fulvestrant with Enzalutamide: 500mg of Fulvestrant will be given IM on days 1, 15, 28, then every 4 weeks as per standard of care (SOC) and 160mg of Enzalutamide will be given PO daily. Patients will receive a tumor biopsy at the start of treatment and 4 weeks after the start of treatment, with an optional 3rd biopsy at the end treatment.

**Secondary Clinical trial ID:**

NCT01830933

INTERVENTION 1:
- Usual Care
- Usual Care is the comparison Clinic Patients, where there is no change in their standard or usual care.

INTERVENTION 2:
- BreastCARE Intervention
- Intervention Clinic Patients: The participants will answer questions on the tablet-PC to calculate their breast cancer risk.
- Intervention Patient Report. Once the patient completes the BreastCare Computer survey, the program will immediately generate a personal feedback report containing information about her risk factors and recommendations to reduce her risk. This report will be printed and given to the patients before she meets with her doctor.
- BreastCARE : The physician will receive a physician report that contains information similar to the patient report.

Using the information above, please write an entailment statement below, please include NCT numbers in the statement. You can use the Annotation guidelines if necessary:

Figure 8: Page 1 of example annotation form.

**Entailment statement:**

**Contradiction statement:**

**Link to Annotation guidelines**

Figure 9: Page 2 of example annotation form.

# 1 Annotation Guidelines

# 2 Motivation

This annotation process is part of a project to build a dataset from clinical trial data. Clinical trials are a type of research that studies new tests and treatments and evaluates their effects on human health outcomes. The clinical trial data in this dataset is breast cancer trials taken from clinicaltrials.gov. The annotations generated through this process will be used for an inference relation task. This will allow further research to develop systems that can evaluate clinical trial texts.

## 2.1 Resources

We have separated the CTRs into 4 sections:

- **Eligibility criteria:** A set of conditions for patients to be allowed to take part in the clinical trial.

- **Intervention:** The type, dosage, frequency, and duration of treatments being studied.

- **Results:** Number of participants in the trial, outcome measures, units, and the results.

- **Adverse Events:** These are signs and symptoms observed in patients during the clinical trial.

| Eligibility Criteria | **Inclusion:**
• HER2 + diagnosis
• 18 years of age or older
**Exclusion:**
• Presence of central nervous system or brain metastases. |
|---|---|
| **Intervention** | **Gemcitabine** 1500 mg/m2 body surface area (BSA) **intravenously** (IV) over 30 minutes (+/- 5 minutes) on days 1 and 15 of each cycle. |
| **Results** | **Variable:** Progression Free Survival
Number of Participants: 29
Median (months): 10.4 (5.6 to 15.2) |
| **Adverse Events** | **Total:** 3/29
• Neutropenic Fever 1 / 29 (3.45%)
• Peripheral Neuropathy 1 / 29 (3.45%)
• Seizure/Syncope 1 / 29 (3.45%) |

You will receive a CTR section prompt, which pertains to one of the four sections, along with the corresponding information from both a primary and secondary trial.

Figure 10: Page 1 of Annotation guidelines.

For every pair of trials, you must write 2 statements, an entailment statement, and a contradictory statement. When writing statements, you must always refer to the primary and secondary trials as **"Primary Clinical Trial"** and **"Secondary Clinical Trial"**. If the data is illegible/erroneous please contact me with the clinical trial ID of the data. We provide you with 100 sample annotations to help demonstrate this process in sample_data.docx.

## 2.2 Entailment statement annotation

Using the information provided write an entailment statement. This statement should make an objectively true claim about the content of the prompted section of the trial(s), you have the option to write a statement about the primary trial alone or to draw a comparison between both the primary and secondary trials. Aim to produce non-trivial statements. This means that to infer the statement's truth, one must read and comprehend multiple rows of the CTR section. Simple techniques for generating Non-trivial statements are summarization, comparisons, negation, relations, inclusion, superlatives, aggregations, and rephrasing. However, this list is not exhaustive. Any statement that demands understanding and reasoning is acceptable. Annotators are encouraged to write non-repetitive statements, that require different types of reasoning, over different types of information. Avoid vague words such as 'maybe' or 'perhaps' etc, that create uncertainty or ambiguity, non-definitive statements will be removed.

## 2.3 Contradictory Statements

Using the entailment statement you've generated, write a contradiction statement. This statement should make an objectively false claim about the content of the prompted section of the trial(s) if the entailment statement is only about the primary trial, the contradiction statement should also only be about the primary trial, conversely if the entailment statement includes both the primary and secondary trials, the contradiction statement should be the same. Where possible aim to maintain the original statement's style and length.

## 2.4 Evidence retrieval

For each statement, annotators are asked to collect evidence as proof of the true/false nature of their statement. Please collect evidence using each line as an individual fact, i.e copy and paste the full line into the evidence section. Multiple lines/facts may be used for evidence. Every statement should have at least one piece of evidence supporting its inference. In instances where negation is used (e.g., "clinical trial A does not have a placebo arm"), you are required to present the entire CTR section as evidence.

Figure 11: Page 2 of Annotation guidelines.