# OpenReview forum: "NLI4CT: Multi-Evidence Natural Language Inference for Clinical Trial Reports"
_EMNLP/2023/Conference — EMNLP 2023 Main_

### Official Review · Reviewer_hG7J · 2023-07-22

**Soundness:** 3

**Excitement:**

4: Strong: This paper deepens the understanding of some phenomenon or lowers the barriers to an existing research direction.

**Paper Topic And Main Contributions:**

The paper introduces a novel binary biomedical NLI benchmark, which utilizes clinical trial reports, specifically focusing on breast cancer. Additionally, it proposes an evidence retrieval task to support the predicted relation (entailment or contradiction). In this paper, each clinical trial report is divided into four sections. These individual sections serve as premises, whereas hypotheses are written by experts. Both tasks provide the means to test the domain knowledge and various capabilities of the models, including numerical and commonsense reasoning. The authors provide several baselines for the proposed benchmark.

**Reasons To Accept:**

* The paper proposes a novel NLI dataset focused on CTRs with an evidence selection task. The task has the potential to contribute to achieving more transparent and explainable predictions, which is highly needed in this domain.

* The paper is easy to follow, with plenty of examples and detailed explanations.

* The categorizations in NLI4CT allow researchers to do more fine-grained analysis (Results Vs. Intervention, Single Vs. Comparison).




**Reasons To Reject:**


* The paper does not report the performance of several recent best-performing biomedical models including PubMedBERT and BioLinkBERT and therefore lack of more domain-specific models evaluation.
* Statements-only baseline revealed the presence of artefacts, which is consistent with existing NLI datasets where hypothesis-only performs well, including MedNLI. It would be beneficial to identify and reduce shortcuts exploited by the models to enhance the overall robustness and reliability of the results.

**Reproducibility:**

4: Could mostly reproduce the results, but there may be some variation because of sample variance or minor variations in their interpretation of the protocol or method.

**Reviewer Confidence:**

4: Quite sure. I tried to check the important points carefully. It's unlikely, though conceivable, that I missed something that should affect my ratings.

**Typos Grammar Style And Presentation Improvements:**


* Inconsistent capitalization in Table 1, such as "neutral".

---

> ### Author Rebuttal · Authors · 2023-08-26
>
> 1. The paper does not report the performance of several recent best-performing biomedical models including PubMedBERT and BioLinkBERT and therefore lack of more domain-specific models evaluation.
>
> The aim of the experiments is to establish a set of baselines for NLI4CT covering different categories of models (including models with different sizes and architectures) and not to exclusively focus on domain-specific models. Regarding this category, we decided to evaluate BioBert and BioMegatron as representative approaches in the research community and leave a more in depth evaluation of domain-specific models for future research. Moreover, it is important to notice that our experiments also cover larger models (e.g. GPT 3.5) that have demonstrated to exceed the performance of bert-based models on a range of previous domain-specific tasks.
>
> 2. Statements-only baseline revealed the presence of artefacts, which is consistent with existing NLI datasets where hypothesis-only performs well, including MedNLI. It would be beneficial to identify and reduce shortcuts exploited by the models to enhance the overall robustness and reliability of the results.
>
> As explained in the paper (Lines 476-488) we investigated several different possible artefacts in our experiments, finding no conclusive evidence of strong statistical biases. Specifically, the statement-only baseline achieves an F1 of 0.626 which is significantly below the majority class baseline (0.66 F1). Since the entailment task is a binary classification task and considering that existing models are well-known for finding spurious correlations in the data, we can conclude that the presence of such artifacts in the dataset is very minimal.

---

### Official Review · Reviewer_6bps · 2023-08-04

**Soundness:** 4

**Ethical Concerns:**

Yes

**Excitement:**

4: Strong: This paper deepens the understanding of some phenomenon or lowers the barriers to an existing research direction.

**Justification For Ethical Concerns:**

Following the Ethics Review questions for papers presenting new datasets, the authors should address these questions in their paper:

1. Describe how intellectual property (copyright, etc) was respected in the data collection process.

2 Describe how crowd workers or other annotators were fairly compensated and how the compensation was determined to be fair.

**Missing References:**

The previous literature review is correct and up to date. The authors highlight the papers that have supported the need for more efficacy of neural models in quantitative reasoning and numerical operations within the NLI.

They also cover in a table the comparison with other similar resources.

References to similar resources on CTR in other languages need to be included. For example:

* Kors JA, Clematide S, Akhondi SA, van Mulligen EM, Rebholz-Schuhmann
D. A multilingual gold-standard corpus for biomedical concept recognition:
the Mantra GSC. J Am Med Inform Assoc. 2015;22(5):948–56.

* Campillos‑Llanos et al. (2021) A clinical trials corpus annotated with UMLS entities to enhance the access to evidence‑based medicine. BMC Med Inform Decis Mak (2021) 21:69 https://doi.org/10.1186/s12911-021-01395-z


(authors would include those references in the final version).

**Paper Topic And Main Contributions:**

The paper falls under "new data resources" and "NLP engineering experiment" topics. It discusses the development of a novel resource called NLI4CT for advancing research on Natural Language Inference (NLI) for reasoning on clinical trial reports (CTRs).

The main hypothesis is that Clinical trial reports (CTR) amassed over the years contain indispensable information for developing personalised medicine. However, it is practically infeasible to manually inspect over 400,000+ clinical trial reports to find the best evidence for experimental treatments. Natural Language Inference (NLI) offers a potential solution to this problem by allowing the scalable computation of textual entailment. However, existing NLI models perform poorly on biomedical corpora, and previously published datasets must capture the full complexity of inference over CTRs.

The contribution of the work is to present a novel resource to advance research on NLI for reasoning on CTRs. The resource applies to two tasks on breast cancer: determining the inference relation between a natural language statement and a CTR and retrieving supporting facts to justify the predicted relation.  They provide a corpus of 2400 expert-annotated instances to train and test models on these tasks. In addition, the paper provides baselines on this corpus to expose the limitations of existing NLI models, with six state-of-the-art NLI models achieving a maximum F1 score of 0.627. The authors claim this is the first attempt to design a task covering the interpretation of full CTRs. They make the corpus, competition leaderboard, website and code available to replicate the baseline experiments.

**Questions For The Authors:**

A. On the annotation process, how many expert annotators were involved, did they have a balanced load on the annotation, how was the annotation guide developed, and what did the annotators encounter the main problems and disagreements?

B. Given that the resource is for NLI model testing only... How much human effort and time would it take to convert this resource into a complete training dataset to generate a real application?

(the answers given in the rebuttal are appropriate).

**Reasons To Accept:**

The most important contributions of the paper:

1. NLI4CT Resource itself. NLP needs good resources annotated by experts.

2. Addressing Biomedical NLI Challenges: CTRs are technical and complex documents challenging natural language inference. The paper highlights the importance of addressing challenges related to numerical reasoning, biomedical NLI, and processing long texts.

3. Baseline Performance Comparison: The paper presents test results with 6 state-of-the-art NLI models on the NLI4CT corpus. These results highlight the current limitations of existing NLI models, especially regarding numerical reasoning. The performance comparison is a benchmark for future improvements and advancements in the field.

4. Availability of Resources: The authors provide public access to their NLI4CT corpus, a competition leaderboard, a website, and the code necessary to replicate the baseline experiments.

Overall, the paper presents contributions to the NLP community in the biomedical field by addressing the challenge of natural language inference in clinical trial reports. The development of the NLI4CT corpus and the findings on the limitations of current models provide a solid basis for future research and improvements in this area. The availability of resources also fosters collaboration and progress in the field.

**Reasons To Reject:**

1. The corpus compilation and its annotation is not fully described. Please see question A below. This point is relevant because the paper should address potential biases and limitations in the annotation process.

2. Potential Biases: The article should address any potential biases introduced during the data collection and annotation. Biases could impact the generalizability and reliability of the findings.

3. The list of limitations acknowledged by the authors: the models developed are not fit for direct medical application; the evaluation also lacks an interventional study to verify if the models learn the underlying causal structure of the tasks; and the dataset contains fewer instances than other published biomedical NLI datasets.

In summary, the main reason for rejecting the paper would be that the scope of the resource is limited to testing NLI models.

(the answers given in the rebuttal are appropriate).

**Reproducibility:**

5: Could easily reproduce the results.

**Reviewer Confidence:**

3: Pretty sure, but there's a chance I missed something. Although I have a good feel for this area in general, I did not carefully check the paper's details, e.g., the math, experimental design, or novelty.

**Typos Grammar Style And Presentation Improvements:**

Overall the paper is well written. There is too much information of different kinds, but it is well distributed in the article's structure.

---

> ### Author Rebuttal · Authors · 2023-08-27
>
> 1. The corpus compilation and its annotation is not fully described. Please see question A below. This point is relevant because the paper should address potential biases and limitations in the annotation process.  A - On the annotation process, how many expert annotators were involved, did they have a balanced load on the annotation, how was the annotation guide developed, and what did the annotators encounter the main problems and disagreements?
>
> In total there were 4 expert annotators, with one expert performing the majority of the annotations. The annotation guide was carefully refined after the annotation of 100 Pilot examples from the main expert annotator to make the whole annotation process as efficient as possible, we will publish the annotation guide in the appendix of the camera ready. There were no significant problems or disagreements throughout the annotation process, as shown in lines 247-251, we achieved a very high rate of agreement (Cohen’s kappa of 0.83) and an 85.66\% average accuracy with respect to the gold labels.
>
>
> 2. Potential Biases: The article should address any potential biases introduced during the data collection and annotation. Biases could impact the generalizability and reliability of the findings.
>
> As explained in the paper (Lines 476-488) we investigated several different possible artefacts/biases in our experiments, finding no conclusive evidence of strong statistical biases. Specifically, the statement-only baseline achieves an F1 of 0.626 which is significantly below the majority class baseline (0.66 F1). Since the entailment task is a binary classification task and considering that existing models are well-known for finding spurious correlations in the data, we can conclude that the presence of such artifacts in the dataset is very minimal.
>
>
> 3. The list of limitations acknowledged by the authors: the models developed are not fit for direct medical application; the evaluation also lacks an interventional study to verify if the models learn the underlying causal structure of the tasks; and the dataset contains fewer instances than other published biomedical NLI datasets.
>
> We believe that developing benchmarks such as NLI4CT represents a necessary step toward the development of models capable of clinical decision-making. However, at the same time, we acknowledge that NLI4CT alone is not enough for real-world applications and it has to be complemented with further analyses, the development of models that are fit for medical applications, generalization studies, and regulatory compliance among other things. Additionally, with respect to the size of our dataset, we address the future steps we will be taking to increase the size of our dataset in the conclusion of our paper.
>
> 4. In summary, the main reason for rejecting the paper would be that the scope of the resource is limited to testing NLI models.
>
> We believe that developing benchmarks such as NLI4CT represents a necessary step toward the development of models capable of clinical decision-making, and this is a rich and complex domain, with enough depth to warrant a paper.  Additionally, we believe that we achieved all of our aims, within this scope.
>
> 5. They also cover in a table the comparison with other similar resources. References to similar resources on CTR in other languages need to be included. For example: Kors JA, Clematide S, Akhondi SA, van Mulligen EM, Rebholz-Schuhmann D. A multilingual gold-standard corpus for biomedical concept recognition: the Mantra GSC. J Am Med Inform Assoc. 2015;22(5):948–56. Campillos‑Llanos et al. (2021) A clinical trials corpus annotated with UMLS entities to enhance the access to evidence‑based medicine. BMC Med Inform Decis Mak (2021) 21:69 https://doi.org/10.1186/s12911-021-01395-z
>
> We will add these references in the related work section of the camera-ready as different tasks in a similar domain (thank you for pointing them out).
>
> 6. Given that the resource is for NLI model testing only... How much human effort and time would it take to convert this resource into a complete training dataset to generate a real application?
>
> We believe that developing benchmarks such as NLI4CT represents a necessary step toward the development of models capable of clinical decision-making. To convert this resource into a complete training dataset to generate a real application NLI4CT has to be complemented with analyses which are focused on their clinical safety, going through a validation process which typically transcends NLP methodologies (i.e. a technology clinical trial). While the size of the dataset that would be required to deliver a model for clinical use is a factor, the most important factors may be not related to the dataset (e.g. quantifying uncertainty). This dataset is a cornerstone enabler towards the construction of these models.
>
> 7. Describe how intellectual property (copyright, etc) was respected in the data collection process. Describe how crowd workers or other annotators were fairly compensated and how the compensation was determined to be fair.
>
> The group of annotators that took part in the annotation process are all authors of the paper and were therefore not compensated. The three annotators for the reclassification were volunteers recruited within the expert network of the authors. These contributed their efforts on a pro bono basis. We will elicit these points in the camera ready.
>
> 8. Describe how intellectual property (copyright, etc) was respected in the data collection process.
>
> To the best of our knowledge, we are entirely in line with the clinicaltrials.gov terms of service, as ClinicalTrials.gov data is available to all requesters, both within and outside the United States In the dataset documentation and in the paper we attribute the source of the data as ClinicalTrials.gov and clearly state the modifications made to the content of the data, along with a complete description of the modifications (Lines 145-172, 199-261). We can elicit these points further in the camera ready.

---

### Official Review · Reviewer_iVg5 · 2023-08-04

**Soundness:** 4

**Excitement:**

4: Strong: This paper deepens the understanding of some phenomenon or lowers the barriers to an existing research direction.

**Paper Topic And Main Contributions:**

This paper defined a novel benchmark, called NLI4CT, for clinical trial reasoning and inference tasks. This benchmark includes two tasks.
This paper also released a new corpus with 2400 expert annotated entailment relations.
This paper tested and compared 7 SOTA NLI models .

**Questions For The Authors:**

Some SOTA LLMs are not tested, for example, GPT-4 or LLaMA.

**Reasons To Accept:**

Extract the information from clinical trials report and inference on the patient eligibility is an important task to facilitate patient recruitment for clinical trials. The problem is very significant for clinical and translational research. The dataset released by this paper will be very valuable to the medical informatics community.

**Reasons To Reject:**

A significant number of literature is missing in the paper. Particularly those from Chunhua Weng (Columbia University)'s group. They have been doing clinical trial information extraction and normalization for decades. This reviewer would like to see how different contributions this work make compared to Weng's group.

**Reproducibility:**

3: Could reproduce the results with some difficulty. The settings of parameters are underspecified or subjectively determined; the training/evaluation data are not widely available.

**Reviewer Confidence:**

5: Positive that my evaluation is correct. I read the paper very carefully and I am very familiar with related work.

---

> ### Author Rebuttal · Authors · 2023-08-26
>
> 1. Questions For The Authors: Some SOTA LLMs are not tested, for example, GPT-4 or LLaMA.
>
> The main focus of the paper is the construction of a new resource for clinical NLI. Therefore, the experiments aim to establish a set of baselines covering different categories of models (including models with different sizes and architectures). Regarding the category of LLMs, we opted for GPT3.5 Turbo since it has a lower cost than GPT4 and it generally exhibits higher overall performance than LLaMa. Moreover, since the focus of the paper is not to exclusively evaluate the performance of LLMs, and these models are very resource intensive, we decided to leave the evaluation of different and possibly larger models for future work. Moreover, our experiments have already shown that specialized pre-training may be more significant than continually increasing model size (Table 3 \& 4).
>
> 2. A significant number of literature is missing in the paper. Particularly those from Chunhua Weng (Columbia University)'s group. They have been doing clinical trial information extraction and normalization for decades. This reviewer would like to see how different contributions this work make compared to Weng's group.
>
> We agree that Weng's group leading contributions on NLP for CTs need to be better compared and contrasted against this contribution. Some more significant differences are around the fact that this work is more centered around a clinical NLI task. Another difference is the focus on this task on inferences beyond eligibility criteria (also including inference over results, adverse events, or intervention sections). We will extend the related work section in the camera ready eliciting the contributions and differences regarding Weng's long standing contributions in NLP for CTs.
>
> 3. Reproducibility: 3: Could reproduce the results with some difficulty. The settings of parameters are underspecified or subjectively determined; the training/evaluation data are not widely available.
>
> Training/evaluation data were submitted in supplementary materials, and links to the data were anonymized, as instructed. Parameter settings are displayed in a table in the appendix (Table 6), and are implemented in a standardized manner.

---

### Meta-Review · Area_Chair_cyjw · 2023-09-22

**Recommendation:** 4

**Metareview:**

As a reviewer indicates this paper defined a novel benchmark, called NLI4CT, for clinical trial reasoning and inference tasks. This benchmark includes two tasks. This paper also released a new corpus with 2400 expert annotated entailment relations. This paper tested and compared 7 SOTA NLI models.

The main reasons to accept the paper are the following ones:


	Extracting the information from clinical trials report and inference on the patient eligibility is an important task to facilitate patient recruitment for clinical trials. The problem is very significant for clinical and translational research. The dataset released by this paper will be very valuable to the medical informatics community.

    NLI4CT Resource itself. NLP needs good resources annotated by experts.

    Addressing Biomedical NLI Challenges: CTRs are technical and complex documents challenging natural language inference. The paper highlights the importance of addressing challenges related to numerical reasoning, biomedical NLI, and processing long texts.

    Baseline Performance Comparison: The paper presents test results with 6 state-of-the-art NLI models on the NLI4CT corpus. These results highlight the current limitations of existing NLI models, especially regarding numerical reasoning. The performance comparison is a benchmark for future improvements and advancements in the field.

    Availability of Resources: The authors provide public access to their NLI4CT corpus, a competition leaderboard, a website, and the code necessary to replicate the baseline experiments.

    The paper is easy to follow, with plenty of examples and detailed explanations.

    The categorizations in NLI4CT allow researchers to do more fine-grained analysis (Results Vs. Intervention, Single Vs. Comparison).


The main reasons to reject the paper are the following ones:


	Some references are missing in the paper.

	The corpus compilation and its annotation is not fully described. The paper should address potential biases and limitations in the annotation process.

    	Potential Biases: The article should address any potential biases introduced during the data collection and annotation. Biases could impact the generalizability and reliability of the findings.

    	The list of limitations acknowledged by the authors: the models developed are not fit for direct medical application; the evaluation also lacks an interventional study to verify if the models learn the underlying causal structure of the tasks; and the dataset contains fewer instances than other published biomedical NLI datasets.

	The paper does not report the performance of several recent best-performing biomedical models including PubMedBERT and BioLinkBERT and therefore lack of more domain-specific models evaluation.

Despite the reasons to reject the paper, reviewers agree that the paper is sound and strong. This resource would help make advances in the field.

If the paper is accepted, the final version should be elaborated taking into account the comment by reviewers and by the ethics committee.

---

### Decision · Program_Chairs · 2023-10-07

**Decision:**

Accept-Main

**Comment:**

As a reviewer indicates this paper defined a novel benchmark, called NLI4CT, for clinical trial reasoning and inference tasks. This benchmark includes two tasks. This paper also released a new corpus with 2400 expert annotated entailment relations. This paper tested and compared 7 SOTA NLI models.

The main reasons to accept the paper are the following ones:


	Extracting the information from clinical trials report and inference on the patient eligibility is an important task to facilitate patient recruitment for clinical trials. The problem is very significant for clinical and translational research. The dataset released by this paper will be very valuable to the medical informatics community.

    NLI4CT Resource itself. NLP needs good resources annotated by experts.

    Addressing Biomedical NLI Challenges: CTRs are technical and complex documents challenging natural language inference. The paper highlights the importance of addressing challenges related to numerical reasoning, biomedical NLI, and processing long texts.

    Baseline Performance Comparison: The paper presents test results with 6 state-of-the-art NLI models on the NLI4CT corpus. These results highlight the current limitations of existing NLI models, especially regarding numerical reasoning. The performance comparison is a benchmark for future improvements and advancements in the field.

    Availability of Resources: The authors provide public access to their NLI4CT corpus, a competition leaderboard, a website, and the code necessary to replicate the baseline experiments.

    The paper is easy to follow, with plenty of examples and detailed explanations.

    The categorizations in NLI4CT allow researchers to do more fine-grained analysis (Results Vs. Intervention, Single Vs. Comparison).


The main reasons to reject the paper are the following ones:


	Some references are missing in the paper.

	The corpus compilation and its annotation is not fully described. The paper should address potential biases and limitations in the annotation process.

    	Potential Biases: The article should address any potential biases introduced during the data collection and annotation. Biases could impact the generalizability and reliability of the findings.

    	The list of limitations acknowledged by the authors: the models developed are not fit for direct medical application; the evaluation also lacks an interventional study to verify if the models learn the underlying causal structure of the tasks; and the dataset contains fewer instances than other published biomedical NLI datasets.

	The paper does not report the performance of several recent best-performing biomedical models including PubMedBERT and BioLinkBERT and therefore lack of more domain-specific models evaluation.

Despite the reasons to reject the paper, reviewers agree that the paper is sound and strong. This resource would help make advances in the field.

If the paper is accepted, the final version should be elaborated taking into account the comment by reviewers and by the ethics committee.